# How to Use the Cuprizone Model to Study De- and Remyelination

**DOI:** 10.3390/ijms25031445

**Published:** 2024-01-24

**Authors:** Markus Kipp

**Affiliations:** Rostock University Medical Center, Institute of Anatomy, 18057 Rostock, Germany; markus.kipp@med.uni-rostock.de; Tel.: +49-(0)-381-494-8400

**Keywords:** myelin, astrocytes, glia, oligodendrocytes, multiple sclerosis, cuprizone, demyelination, remyelination, protocol, guideline

## Abstract

Multiple sclerosis (MS) is an autoimmune and inflammatory disorder affecting the central nervous system whose cause is still largely unknown. Oligodendrocyte degeneration results in demyelination of axons, which can eventually be repaired by a mechanism called remyelination. Prevention of demyelination and the pharmacological support of remyelination are two promising strategies to ameliorate disease progression in MS patients. The cuprizone model is commonly employed to investigate oligodendrocyte degeneration mechanisms or to explore remyelination pathways. During the last decades, several different protocols have been applied, and all have their pros and cons. This article intends to offer guidance for conducting pre-clinical trials using the cuprizone model in mice, focusing on discovering new treatment approaches to prevent oligodendrocyte degeneration or enhance remyelination.

## 1. Introduction

The central nervous system (CNS) comprises neurons and neuroglial cells. Neurons are responsible for receiving and transmitting nerve impulses across their cell membranes to adjacent neurons, creating a highly efficient communication network. Critical to this process is the presence of myelin sheaths formed by the expanded and altered plasma membranes of a specific type of glial cells known as oligodendrocytes. These sheaths are essential for the rapid and energy-efficient propagation of action potentials [1]. A wide and increasing variety of neurological human disorders are linked to CNS myelin abnormalities. This group includes the well-known myelin disorder multiple sclerosis (MS) but extends to rare genetic diseases, such as Niemann-Pick disease. Importantly, dysfunctions in oligodendrocytes are also implicated in various neurodegenerative and psychiatric conditions, including Alzheimer’s and Parkinson’s disease, eating disorders, depression, or schizophrenia [2,3,4,5,6].

MS is an autoimmune and inflammatory disorder affecting the CNS, whose cause is still largely unknown. There are three primary clinical courses of MS: relapsing-remitting, secondary progressive, and primary progressive [7]. Relapsing-remitting MS (RRMS) is characterized by episodes of distinct clinical deterioration (i.e., relapses), followed by periods where patients present with stable clinical disease (i.e., remission). Often, RRMS develops into a secondary progressive course (SPMS) after about 10–15 years. In this later disease stage, individual relapses become less frequent, but there is a gradual, sometimes steady, worsening of clinical symptoms. In primary progressive MS (PPMS), the start of the disease is not characterized by distinct relapses but, rather, by a progressive accumulation of clinical disability. The main histopathological characteristics of MS include the infiltration of peripheral immune cells (mainly monocytes and lymphocytes), the compromise of blood-brain-barrier integrity, reactive gliosis, neuroaxonal degeneration, damage to oligodendrocytes, and, most critically, demyelination. Mechanistically, the interaction of autoimmune T- and B-lymphocytes with myelin antigens is believed to trigger the disruption of myelin’s function [8]. Moreover, intrinsic cytodegenerative events in the brain play an essential role in shaping the immune response [9] and triggering peripheral immune cell recruitment [10]. To study the distinct aspects of MS pathology, several different animal models exist, including the experimental autoimmune encephalomyelitis (EAE) model, Theiler’s Murine Encephalomyelitis Virus (TMEV) model, the lysophosphatidylcholine (LPC or lysolecithin) model, and the cuprizone model [11,12,13,14,15,16].

In the cuprizone model, feeding young mice (around 8 weeks old) with cuprizone induces early and selective oligodendrocyte stress, which is believed to involve mitochondrial disruption due to a cuprizone-induced copper deficit, ferroptosis [17], and endoplasmic reticulum stress responses [18]. The exact mechanisms of oligodendrocyte degeneration remain unclear. However, emerging research indicates that the detrimental effects of cuprizone may not be solely caused by copper chelation or a selective impact on oligodendrocytes. Instead, it has been proposed that a reactive cuprizone–copper complex is responsible for the toxic effects of cuprizone and that various cell types within the CNS are affected [19,20,21]. Why cuprizone predominately compromises oligodendrocyte cell metabolism is equally unknown, but the energy-intensive nature of myelin synthesis and maintenance might play a key role. After 5–6 weeks of cuprizone exposure, there is complete demyelination of distinct subregions (called acute demyelination), paralleled by extensive microglial and astrocytic proliferation and damage to axons [22,23,24,25]. If the animals are provided standard chow after acute cuprizone-induced demyelination, endogenous remyelination occurs, which is on the histological level complete after 3–4 weeks [25]. In contrast to acute demyelination, remyelination is highly impaired/delayed when cuprizone administration is prolonged (12–13 weeks or even longer), a process called ‘‘chronic demyelination’’ [26,27,28,29,30] (see Figure 1). Although the primary region of interest in many studies is the corpus callosum (see later in this manuscript), cuprizone-induced demyelination affects multiple white and grey matter brain areas [31].

The cuprizone model is commonly employed to investigate oligodendrocyte degeneration mechanisms or explore remyelination pathways. This article intends to offer guidance for conducting pre-clinical trials using the cuprizone model in mice, focusing on discovering new treatment approaches to prevent oligodendrocyte degeneration or enhance remyelination.

## 2. Oligodendrocyte Degeneration

Cell death, a fundamental biological process, plays a crucial role in multicellular organisms’ development, homeostasis, and pathophysiology. Contrary to the traditional view that categorizes cell death as accidental or controlled [34], recent advances in molecular biology have revealed a broad spectrum of cell death modalities, each with unique characteristics and implications. Often termed “programmed cell death”, apoptosis is a tightly regulated and energy-dependent process characterized by cell shrinkage, nuclear fragmentation, and membrane blebbing. Driven by a cascade of cysteine proteases called caspases, apoptosis ensures the orderly removal of superfluous or damaged cells without provoking an inflammatory response [35,36,37]. Apoptosis is crucial for embryogenesis, tissue remodeling, and immune system function. In contrast, necrosis is traditionally viewed as an uncontrolled form of cell death resulting from acute injury or trauma. Necrosis is marked by cellular swelling, a loss of membrane integrity, and the eventual rupture of the entire cell. During this process, intracellular components are released into the surrounding tissue, triggering inflammation. Several other cell death mechanisms have been discovered and described in recent years: necroptosis [38,39], regulated by receptor-interacting protein kinases (RIPK1 and RIPK3); caspase-1-driven pyroptosis [40]; autophagy, a survival mechanism wherein cells degrade their components in response to stressors, such as nutrient deprivation, but can culminate in cell death; or ferroptosis [41], which is characterized, among others, by lipid peroxidation due to iron-dependent accumulation of reactive oxygen species. A reduced mitochondrial size and an increased mitochondrial membrane density characterize ferroptosis. It has been associated with various pathological conditions, including cancer and neurodegenerative diseases [42,43,44]. A nuanced understanding of these different types of cell death offers insights into cellular homeostasis and development and provides potential therapeutic targets for a myriad of diseases, including MS.

As pointed out above, one of the early outcomes of cuprizone exposure is the selective degeneration of mature oligodendrocytes. Different techniques have been applied to visualize degenerating oligodendrocytes in the cuprizone model, including TdT-mediated dUTP nick-end labeling (better known as TUNEL-labelling) [45,46,47], visualization of the expression of active caspase-3 [48], or the detection of apoptotic bodies in H&E-stained sections [22]. While numerous studies suggest that oligodendrocytes mainly degenerate via apoptosis, the results of a recent study suggest that, besides apoptosis, ferroptosis plays a functional role during oligodendrocyte death [17]. Oligodendrocyte degeneration is an early and specific event in the cuprizone model. Numerous degenerating oligodendrocytes (i.e., condensed and/or fragmented nuclei of cells in a chain-like formation) can be seen in the corpus callosum as early as 2 days after initiation of the cuprizone intoxication. In contrast, other glial cells, such as glial fibrillary acidic protein (GFAP)^+^ astrocytes or IBA1^+^ microglia, do not show any signs of cell death [22,45]. Oligodendrocyte degeneration is paralleled by a pronounced reduction in cells staining positive for mature oligodendrocyte marker proteins, such as Adenomatous-polyposis-coli protein (APC), which is recognized by the CC1 antibody, Glutathione S-Transferase π (GSTπ), or Nogo-A [22,45,49]. In a recent study conducted by Samuel David and colleagues, cuprizone intoxication induced a rapid loss of ~65% of mature oligodendrocytes by 2 days and 80% loss by 1 week after the onset of the cuprizone intoxication [17].

One promising strategy for ameliorating MS disease progression is the protection of oligodendrocytes. Due to the selective degeneration of these cells during early cuprizone intoxication, this model is particularly suitable for studying pathways and drugs able to ameliorate oligodendrocyte stress and in consequence death. In this section, points that might be considered during the experimental design of a study addressing this point are discussed.

A typical experimental setup to study the potential effectiveness of drugs to ameliorate oligodendrocyte degeneration would last up to 1 week, and we suggest including an early (id est, 3 days) and more advanced (id est, 7 days) time point (see Figure 2, and compare with Figure 1).

To evaluate the extent of oligodendrocyte loss, one might label the dying cell population or visualize and quantify the remaining mature oligodendrocyte population using, for example, anti-APC, anti-GSTπ, or anti-Nogo-A immunohistochemistry [50]. Since it has been suggested that some APC^+^ cells might as well be astrocytes or glial progenitor cells [51], anti-OLIG2/APC double-labeling experiments might be performed to more specifically label mature oligodendrocytes, which should be positive for both marker proteins. One should note that the transcription factor OLIG2 is expressed by both, oligodendrocyte progenitor cells and mature oligodendrocytes. Single labeling with anti-OLIG2 antibodies, thus, provides limited information about mature oligodendrocyte densities. The visualization of stress-related proteins, such as DNA damage-inducible transcript 3 (DDIT3) [18], activating transcription factor 3 (ATF3) [48], or 8-Hydroxy-2′-deoxyguanosine (8-OHdG) [52], might be an additional option for estimating the degree of early oligodendrocyte stress and, hence, degeneration. To estimate the extent of early myelin damage, the visualization of major myelin proteins, such as proteolipid protein (PLP) or myelin basic protein (MBP), is not considered to be an appropriate strategy because no overt differences can be observed at this early time point [22,45,53]. Instead, ultrastructural studies [54] or Nile Red spectroscopy [17] might be applied. Of note, our group recently demonstrated that the expression of the ankyrin G and contactin-associated proteins, essential nodal and paranodal proteins, is dramatically reduced after 1 week of cuprizone intoxication [54], suggesting that early oligodendrocyte degeneration is linked to myelin damage in the cuprizone model. Finally, since oligodendrocyte degeneration is closely paralleled by microglia activation, the analysis of microglia densities and/or morphology is an excellent indirect indicator for the extent of oligodendrocyte degeneration [55].

From a functional point of view, myelinogenesis and maintenance are highly complex processes that require cell cycle exit and extensive differentiation processes of oligodendrocyte progenitor cells. According to estimates from morphometric analyses, the mean surface area of myelin per mature myelinating cell is thousands of times greater than the surface area of a typical mammalian cell [56]. To form and maintain the myelin sheath, oligodendrocytes synthesize large amounts of lipids and proteins within the endoplasmic reticulum (ER), increasing their sensitivity to the secretory pathway perturbations. Due to the turnover of myelin lipids and proteins, oligodendrocyte sensitivity to extrinsic and intrinsic disturbances persists until maturity. Recent studies indicate that this extraordinary oligodendrocyte susceptibility might contribute to the pathogenesis of several myelin disorders, including MS [56]. A plethora of cell stress conditions can perturb the fine-tuned folding machinery within the ER, a phenomenon referred to as ER stress. Consequently, cells have evolved an adaptive coordinated response to limit the accumulation of unfolded proteins in the ER: the unfolded protein response (UPR). In higher eukaryotes, the UPR is activated by three ER transmembrane sensors: protein kinase RNA-like endoplasmic reticulum kinase (PERK), activating transcription factor 6 (ATF6), and inositol-requiring enzyme-1 (IRE1) [57]. Mechanistically, the UPR can relieve cell stress by (i) downregulating translation and, thus, decreasing protein load within the ER-lumen, (ii) up-regulating genes encoding ER chaperones and enzymes to increase the ER folding capacity, and (iii) inducing pathways to enhance the degradation of misfolded proteins from the ER. However, if cell stress persists and protective mechanisms fail, apoptosis is induced, among others, via the transcription factor Chop (C/EBP homologous protein), also known as DDIT3. Studies using *Ddit3^−/−^* mice have established the role of DDIT3 during stress-induced apoptosis in several disease models, including renal dysfunction [58], diabetes [59], ethanol-induced hepatocyte injury [60], experimental colitis [61], advanced atherosclerosis [62] or cardiac-pressure overload [63]. Furthermore, there is strong evidence that DDIT3 regulates cell death in neurodegenerative and neuroinflammatory disorders, including Parkinson’s disease [64], subarachnoid hemorrhage [65], Alzheimer’s disease [66], multiple system atrophy [67], and spinal cord injury [68]. Notably, various ER stress markers are expressed in inflammatory MS lesions [69,70], as well as in experimental autoimmune encephalomyelitis (EAE) [71,72] and the cuprizone model [48]. Notably, oligodendrocyte degeneration and demyelination are ameliorated in *Ddit3^−/−^* compared with wild-type mice [18].

## 3. Demyelination

It is well known that continuous intoxication with cuprizone induces demyelination of various brain regions among the white matter tract corpus callosum. As an essential white matter tract of the CNS, the human corpus callosum comprises approximately 150–190 million myelinated nerve fibers that form homotopic or heterotopic projections to contralateral neurons in the same anatomical layer [73,74]. In the sagittal plane, the corpus callosum can be divided into four sections: the rostrum, a curved, front portion bending downward and backward; the genu, an anterior, thick segment bending back to form a knee-like protrusion; the body or trunk, the central part; and the splenium, a posterior, thick section creating a bulbous mass. In the frontal plane, a medial section, flanked on both sites by lateral parts, can be divided.

Some important aspects should be considered when using the cuprizone model to investigate the potentially protective properties of substances on demyelination. First, it is important to note, especially for accurate comparative histological assessments, that demyelination in the corpus callosum is not equally distributed in this model. For instance, in the rostral part of the corpus callosum, the demyelination is extensive and nearly complete in the lateral areas, but it is patchy and less severe along the midline. In contrast, at more caudal levels in the body of the corpus callosum, the lateral areas show less demyelination, while demyelination is severe and consistent in the midline (see arrowheads and star in Figure 3A). This fact necessitates comparing similar brain levels across different experimental animals [75]. In our laboratory, we focus on **two** specific brain regions which are clearly distinguishable in coronal brain sections. The first region is at the level of the anterior commissure (identified as slide 53 in the Allen Brain Atlas or slide 215 in Sidmann et al.’s High Resolution Mouse Brain Atlas [76]), where the olfactory limbs converge at the brain’s midline. This convergence creates a distinct demarcation in the mouse brain (see Figure 3). At this juncture, we recommend separate analyses of the medial and lateral aspects of the genu part of the corpus callosum, delineated by the cingulum’s borders (see dashed line in Figure 3A). The second region is located at the level of the rostral hippocampus (slide 64 in the Allen Brain Atlas or slide 265 in Sidmann et al.’s Atlas [76]), where the hippocampal cornu ammonis region’s pyramidal layer is just becoming visible. Here, we advise focusing on the midline of the trunk part of the corpus callosum for analysis. The spatial arrangement of the corpus callosum and its adjacent structures are equally crucial for precise histological examinations of brain sections. Although the medial portions of the corpus callosum experience significant demyelination, the decrease in myelin staining intensity is much less pronounced in the fornix, the white matter tract lying beneath it [77].

Second, potential protective drug effects are best studied using an experimental setting where demyelination is semimaximal and incomplete. However, after 5–6 weeks of cuprizone intoxication, lateral parts of the corpus callosum genu and medial parts of the corpus callosum trunk are, as outlined above, almost entirely demyelinated, potentially limiting protective drug effects.

Three strategies could be employed to enhance the experimental conditions for this aspect. First, one might focus on those regions where demyelination is incomplete, such as the midline of the corpus callosum genu. This is illustrated in Figure 3A. While after 5 weeks of cuprizone intoxication, the midline of the corpus callosum trunk is almost completely demyelinated (right anti-PLP stained images, highlighted by the star), demyelination is incomplete in the midline of the corpus callosum genu (left anti-PLP stained images). Second, one might terminate the experiment after 3 weeks and analyze the tissue while myelin damage and removal are ongoing and not at their peak. This is illustrated in Figure 3B. While myelin disintegration is visible in LFB/PAS-stained sections, anti-PLP staining intensity loss is less obvious. Third, one might rely on the observation that 3 weeks of cuprizone intoxication is enough to allow for autonomous lesion progression until week 5 (not illustrated in Figure 3) [10,78]. Regarding the second option, we investigated in a recent paper the impact of the transcription factor DDIT3 on cuprizone-induced oligodendrocyte degeneration and demyelination (the relevance of DDIT3 in the context of ER-stress has been outlined already above). To this end, *Ddit3^−/−^* and wild-type mice were intoxicated using cuprizone for either 1 or 3 weeks, and different cellular parameters were assessed. Among other cellular parameters, we observed lower densities of degenerating oligodendrocytes in *Ddit3^−/−^* compared with wild-type mice, paralleled by less pronounced early microglia activation and chemokine mRNA expression. At week 3, we observed preservation of LFB-staining intensities, less accumulation of microglia and macrophages, and less severe acute axonal injury [18]. These results suggest that the unfolded protein response-related transcription factor DDIT3 regulates oligodendrocyte apoptosis and subsequent demyelination in the cuprizone model. Of note, follow-up experiments revealed that the extent of demyelination is comparable in *Ddit3^−/−^* and wild-type mice after 5 weeks of continuous cuprizone intoxication, underpinning that protective effects might be lost in case demyelination induction is too severe. Regarding the third option (id est, lesion progression as an experimental paradigm), in a recently published study, we demonstrated that siponimod, a pharmacological modulator of sphingosin-1-phosphate receptors, ameliorates cuprizone-induced demyelination. In this study, mice were intoxicated using cuprizone for 3 weeks (±siponimod treatment) and afterwards provided standard chow for another 2 weeks (without siponimod treatment) to allow for autonomous lesion progression [79]. Among others, we found amelioration of LFB and anti-PLP staining intensity loss, less severe glia activation, and less severe acute axonal injury. Of note, similar protective effects were found when animals were sacrificed at the end of the 3 weeks cuprizone intoxication period, underpinning the validity of both strategies.

It should be noted at this point that the gold standard to determine the extent of demyelination in different experimental groups would be electron microscopy. While this technique is highly time-consuming, we and others usually rely on more accessible and faster to-conduct staining methods, such as the histochemical luxol-fast-blue (LFB) stain or immunohistological techniques visualizing the expression of different myelin-related proteins, such as PLP, MBP, or MAG. All of these measures are only an estimate of the extent of demyelination. Of note, histochemical stains, such as the LFB stain, visualizing complex and non-solvent extractable phospholipids [79,80,81,82,83,84], are more sensitive to detecting initial myelin sheath damage (see Figure 3B). Consequently, if experiments are terminated after week 3, myelination levels of the corpus callosum are best estimated with histochemical stains.

## 4. Remyelination

Demyelination remains a crucial aspect of MS pathology. However, recent discoveries highlight that significant neuronal damage is just as crucial. Remarkably, remyelination is a prime example of tissue repair in the human CNS. While regenerating destroyed neurons and their axons in the adult CNS is minimal, repairing lost myelin sheaths is principally possible. About 20–30% of examined postmortem MS tissues show signs of remyelination, which can happen at both early and late stages of the disease [85,86]. The remyelination process involves the activation and proliferation of oligodendrocyte progenitor cells (OPCs), their migration toward demyelinated axons, and the interaction with these axons, leading to OPC differentiation and remyelination. Notably, the OPCs can originate from different sources, including progenitor cells scattered in the brain tissue or situated in specific neurogenic niches. The positive effects of remyelination are widely recognized, including the reinstatement of impaired or lost axonal conduction [87], the provision of axonal protection, and functional recovery [88]. A vital aspect of this protection is believed to stem from the nutritional support glial cells provide to axons [89,90]. Moreover, remyelination strongly reduces energy expenditure by reducing the high ATP consumption needed to restore ion gradients through the Na+, K+-ATPase and, thus, can be neuroprotective.

Conceptually, remyelinating cells originate from three primary sources. The first source is neural stem cells (NSCs), found in the subventricular zone (SVZ). These cells can migrate to areas such as the corpus callosum, striatum, and fimbria, where they differentiate into either NG2-positive non-myelinating or mature myelinating oligodendrocytes [91,92,93]. It has been demonstrated that in the cuprizone model, SVZ-NSCs are recruited during the remyelination phase to the white matter tract corpus callosum and can form new myelinating oligodendrocytes [91]. When these SVZ-derived NSCs are ablated, animals display reduced oligodendrocyte numbers within the lesioned corpus callosum [94]. Subgranular zone-derived stem cells mainly differentiate into neurons unless they are genetically reprogrammed and/or trophically manipulated to produce oligodendrocytes [95]. Another cell type that can give rise to new myelinating oligodendrocytes is the OPC. These OPCs are widely distributed across both the white and grey matter of the CNS [96,97,98,99] and therefore called parenchymal OPCs (pOPCs). Following myelin damage, OPCs quickly proliferate and transform into remyelinating oligodendrocytes, playing a pivotal role in myelin restoration [93,100]. Additionally, findings from some studies suggest that adult oligodendrocytes, which withstand the demyelinating event, may also contribute to myelin repair [101,102]. Moreover, the results of a recent study revealed that the median eminence of the hypothalamus might also represent a site for myelin-producing cells [103].

Experiments using the cuprizone model have provided essential insights into the physiology and pathology of remyelination. Among others, this model has been used to demonstrate the importance of vasculature-associated oligodendrocytes [104], investigate the relevance of T-cells for de- and remyelination [105], measure changes of visual evoked potentials during de- and remyelination [106], and understand to what extent de- and remyelination can be assessed using advanced imaging techniques such as ultrahigh-field multiparameter MRI [107].

A sine qua non to study remyelination is a consistent demyelinating event. Thus, remyelination can best be studied in models where demyelination occurs at a predictable anatomical site and follows a well-defined and reproducible kinetic. Both are characteristics of the cuprizone model.

In planning an experiment to investigate whether a substance or a specific protein affects endogenous remyelination, one should consider several essential characteristics of the model. As mentioned, if animals are intoxicated using cuprizone for 5–6 weeks and provided standard chow after that, spontaneous endogenous remyelination occurs during the subsequent weeks. In contrast, remyelination is impaired after prolonged, chronic cuprizone exposure. Of note, any substance or drug applied after acute cuprizone-induced demyelination might well accelerate this active endogenous repair process, but just limited conclusion can be drawn regarding induction of remyelination in the non-supportive environment. Alternatively, one might consider testing the substance after chronic cuprizone-induced demyelination, where endogenous remyelination is impaired.

The reason for remyelination failure after chronic cuprizone-induced demyelination is still unknown. During 12 weeks of continuous cuprizone intoxication protocol, it has been shown that GSTπ+ mature oligodendrocyte densities are significantly reduced until week 5 but then recover around week 6. Between weeks 6 and 12, these maturating oligodendrocytes become again vulnerable against the cuprizone intoxication, resulting in a second wave of oligodendrocyte apoptosis, detected using the TUNEL assay [28]. This second wave of mature oligodendrocyte apoptosis is paralleled by a progressive loss of NG2^+^ progenitors within the chronically demyelinated corpus callosum, which might explain the impaired endogenous remyelination capacity after chronic cuprizone-induced demyelination. Indeed, long-term feeding of cuprizone induces an episodic sequence of demyelinating and remyelinating events, which becomes ineffective over time, progressing to a chronic state of demyelination [26]. The relevance of axonal injury for remyelination failure after chronic demyelination is controversially discussed. Mason and colleagues rarely observed necrotic axons but showed a mean decrease in axonal caliber, indicating some kind of axonal injury or dysfunction that might impair remyelination [26]. In line with this observation, we and others showed a disturbed axonal transport machinery in cuprizone-intoxicated mice, evidenced by the accumulation of pre-synaptic proteins and/or mitochondria [29,108,109,110,111]. Alternatively, alterations in the growth factor environment, which are not permitted for oligodendrocyte progenitor proliferation and differentiation after chronic cuprizone-induced demyelination, might as well play a role. 

Treatment duration is a second important aspect that should be considered. To be able to see positive drug effects, treatment duration should be long enough to allow a drug to activate pro-myelinating pathways; however, if treatment is too long, remyelination might be progressed too far to allow sufficient analysis of pro-myelinating drug effects. For example, it has been shown that 4 weeks after acute cuprizone-induced demyelination, the densities of GSTπ^+^ mature oligodendrocytes return to control levels [112]. Thus, shorter treatment periods should be chosen. After a 5 weeks cuprizone intoxication period, we usually allow the mice to remyelinate for 1 and 2 weeks before termination of the in vivo experimental phase (see Figure 4).

Another critical question regarding a promising trial design is the time point when drug administration should be started. Demyelination and the subsequent remyelination of CNS axons are a complicated, multi-step process involving several different cell types. After cuprizone-induced oligodendrocyte injury, microglia and astrocytes are activated by still-unknown signals, leading to myelin disintegration, phagocytosis of myelin debris, and, finally, demyelination. It has been demonstrated that OPCs first begin to accumulate in the corpus callosum around 2–3 weeks after initiating the cuprizone intoxication [31,112,113,114]. If one expects that the applied drug might induce the proliferation rather than the differentiation of OPCs (or eventually both), drug treatment should be started around week 3 and eventually continued until the end of the experiment.

One should also consider the potential source of the remyelinating cells. As pointed out above, remyelinating oligodendrocytes might principally originate from SVZ-derived NPCs or parenchymal OPCs (pOPCs), the latter being widely distributed in the white and grey matter parenchyma [96,97,98,99] and defined by the expression of platelet-derived growth factor receptorα (PDGFRα) and NG2 proteoglycan. Xing and colleagues investigated the extent to which NPCs, as opposed to pOPCs, contribute to remyelination in the cuprizone model. They applied CreER-loxP fate mapping of NPCs and pOPCs to directly assess the relative contribution of these distinct precursor populations to oligodendrogenesis and remyelination after cuprizone-induced demyelination of the corpus callosum. They showed that NPC-derived oligodendrocytes are an important source of new oligodendrocytes in regions of the rostral but not the caudal corpus callosum [115]. Keeping this in mind, if substances are used that are expected to modulate NPC-derived oligodendrocytes rather than pOPC, such as Gli1 inhibitors [116], the focus of the region of interest should be at rostral rather than caudal levels of the corpus callosum. This strategy increases the chance to see pro-myelinating effects of NPC-derived oligodendrocytes.

## 5. Conclusions

The cuprizone model is a simple yet effective tool for studying various elements of MS pathology, including oligodendrocyte degeneration, demyelination, and remyelination. In contrast to the EAE model, cuprizone-induced demyelination does not lead to severe motor deficits [117], limiting the evaluation of motor behavior as functional readout. While ultrastructural studies are the gold standard for evaluating myelin integrity and de- and remyelination, electrophysiological studies represent a powerful tool for assessing the functional status of the axon-myelin unit [118]. However, systematic studies addressing changes of electrophysiological properties following de- and remyelination in this model are largely missing. Crawford and colleagues were able to demonstrate that cuprizone intoxication for 3 to 6 weeks induced significant demyelination in the corpus callosum, decreased axonal conduction, and caused deterioration in axonal structural integrity, including at the nodes of Ranvier. Substituting the cuprizone diet with a regular diet resulted in myelin regeneration; however, it did not completely restore the deficits in conduction and structure [119]. A decrease in axonal conduction during cuprizone-induced demyelination was also found by our group [54]. Visual evoked potentials [120], a measurement frequently used during remyelination trials in humans, might be another promising additional readout not only in the EAE but also in the cuprizone model [106]. However, the extent of demyelination and axonal injury in the visual pathway of cuprizone-intoxicated mice remains to be investigated. While there are proteomic alterations, clear demyelination was not seen in one study using a low-dose (id est, 0.1%) cuprizone intoxication protocol [121]. 

One limitation of the cuprizone model is that the mode of cuprizone-induced oligodendrocyte degeneration still needs to be fully elucidated. This fact limits the translational validity of the model. However, there have been significant advancements in this field in recent years. In 2022, a copper-dependent regulated cell death resulting from direct binding of copper to lipoylated components of the tricarboxylic acid was described [122]. The relevance of this newly-termed cell death modality, known as “cuproptosis” [123], for MS in general and cuprizone-induced oligodendrocyte degeneration, in particular, remains to be clarified in future studies.

Despite this limitation, the cuprizone model is a powerful tool for developing novel therapeutic options in MS and other myelin-related disorders. Changes to only a few key elements can optimize the use of the cuprizone model for addressing specific scientific questions. We hope that this article will assist in improving the design of pre-clinical studies and optimize experimental outcomes.

## Figures and Tables

**Figure 1 ijms-25-01445-f001:**
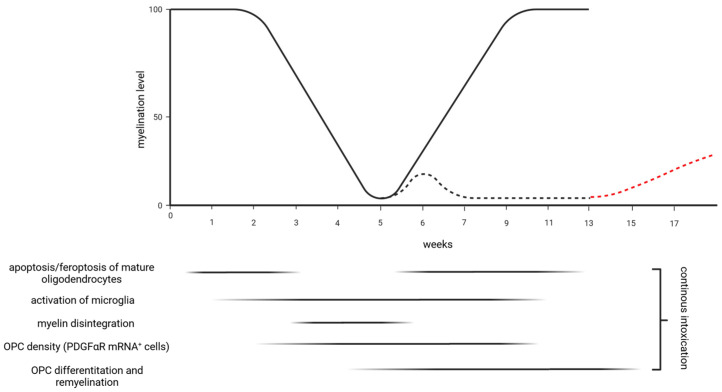
Schematic illustration of cellular key events during cuprizone-induced demyelination (0.2–0.25%) and subsequent remyelination if animals are provided standard chow. After 5 weeks of cuprizone intoxication, demyelination of the midline of the corpus callosum at the level of the rostral hippocampus is almost complete (id est, acute demyelination). Robust endogenous remyelination occurs if the animals are provided standard chow after acute demyelination. Note that the velocity of remyelination is overestimated in case remyelination is assessed via histochemistry compared with electron-microscopy [26]. After 13 weeks of cuprizone intoxication, demyelination of the midline of the corpus callosum at the level of the rostral hippocampus is still almost complete (id est, chronic demyelination; dashed line). However, endogenous remyelination is impaired after chronic cuprizone-induced demyelination (red dashed line). The lower part of the image shows cellular characteristics during a continuous cuprizone intoxication. OPC (oligodendrocyte progenitor cells); PDGFα (platelet-derived growth factor receptorα). The figure is adapted from [32,33].

**Figure 2 ijms-25-01445-f002:**
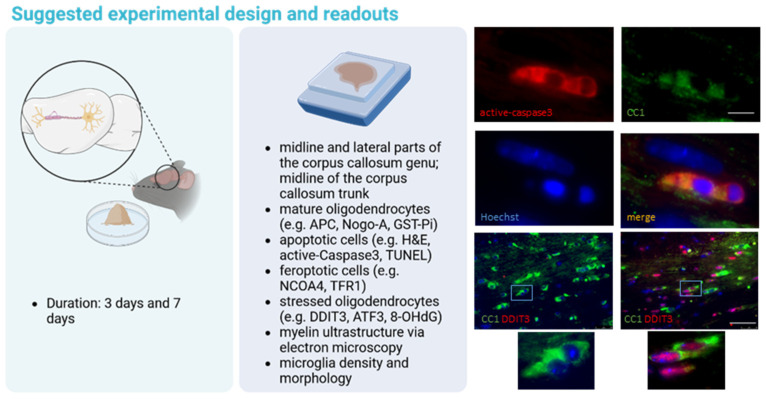
Schematic illustration demonstrating a typical experimental setup to study the potential effectiveness of drugs to ameliorate oligodendrocyte degeneration. The left image illustrates the duration of the cuprizone intoxication. The central image lists the recommended readouts for studying oligodendrocyte densities, cell stress, myelination, and glial reactions. On the right, representative images of immunofluorescence double-labeling experiments are shown. They depict an apoptotic APC/CC1^+^ mature oligodendrocyte expressing active-caspase-3 (upper part) and a stressed APC/CC1^+^ mature oligodendrocyte expressing the transcription factor DDIT3. APC (adenomatous polyposis coli); GST (glutathionine S-transferase); TUNEL (TdT-mediated dUTP-biotin nick end labeling); NCOA4 (nuclear receptor coactivator 4); TFR1 (transferrin receptor 1); DDIT3 (DNA damage-inducible transcript 3); ATF3 (activating transcription factor 3); 8-OHdG (8-Hydroxy-2′-deoxyguanosine). Scale bar 10 µm (upper part) and 80 µm (lowert part). Created with https://www.biorender.com at the 10 January 2024.

**Figure 3 ijms-25-01445-f003:**
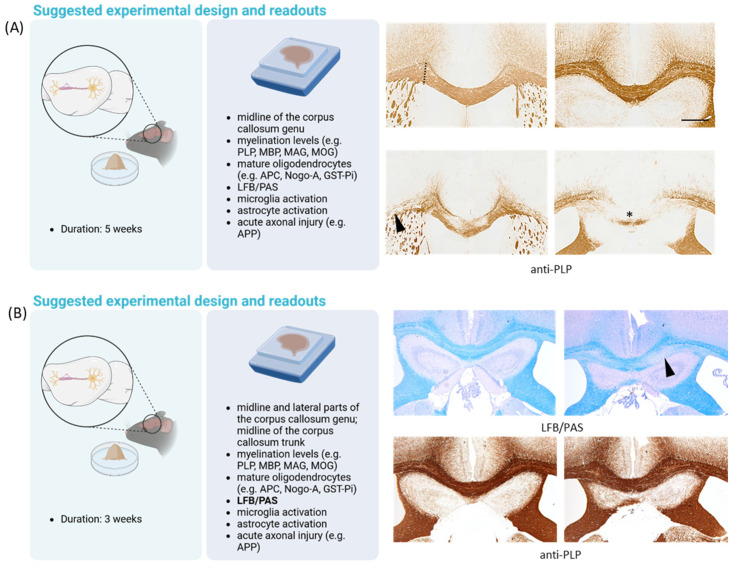
Schematic illustration demonstrating a typical experimental setup for studying the potential effectiveness of drugs to ameliorate cuprizone-induced demyelination. In (**A**), mice are intoxicated for 5 weeks, whereas in (**B**), mice are intoxicated for 3 weeks (0.25%). The left images in (**A**) and (**B**) illustrate the duration of the cuprizone intoxication. The central images in (**A**,**B**) list the recommended readouts for studying myelination levels, oligodendrocyte densities, glial reactions, and acute axonal injury. On the right image in (**A**), representative images of anti-PLP stained sections highlight the region-specific demyelination of the corpus callosum. The arrowhead in (**A**) highlights the demyelinated lateral part of the corpus callous genu, whereas the star in (**A**) highlights the demyelinated medial part of the corpus callosum trunk. Note that the fornix, which lies directly beneath the midline of the corpus callosum, is less affected by cuprizone. On the right image in (**B**), representative LFB/PAS and anti-PLP stained sections highlight the relative sensitivity of both methods to visualize early myelin degeneration. The arrowhead in (**B**) highlights partial demyelination of the corpus callosum trunk visible in LFB/PAS but not anti-PLP processed sections. PLP (proteolipid protein 1); MBP (myelin basic protein); MAG (myelin-associated glycoprotein); MOG (myelin oligodendrocyte glycoprotein); APC (adenomatous polyposis coli); GST (glutathionine S-transferase); LFB/PAS (luxol-fast-blue/periodic acid–Schiff reaction); APP (amyloid beta precursor protein). Scale bar 250 µm. Created with https://www.biorender.com at the 10 January 2024.

**Figure 4 ijms-25-01445-f004:**
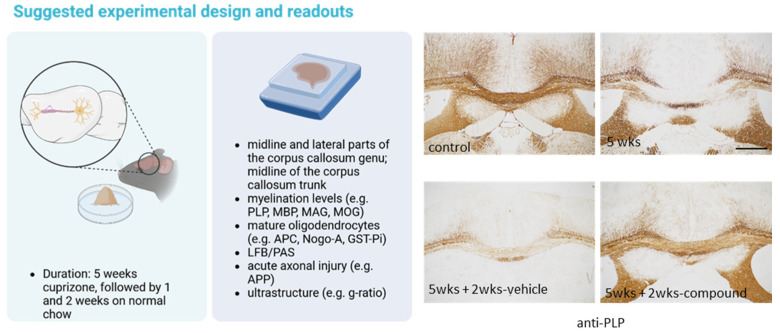
Schematic illustration demonstrating an experimental setup for studying the potential effectiveness of drugs to support remyelination. The left image illustrates the duration of the cuprizone intoxication (0.25%) and subsequent remyelination. The central image lists the recommended readouts for studying myelination levels, oligodendrocyte densities, and acute axonal injury. On the right site, representative anti-PLP processed sections highlight demyelination at week 5 and partial remyelination after 2 weeks on standard chow. Note that after 2 weeks of recovery, remyelination is incomplete and significantly boosted by compound treatment. Further note that by using electron microscopy, a so-called ‘g-ratio’ can be calculated by dividing the diameter of the axon by the diameter of the axon+myelin sheath. Since remyelinated fibers are characterized by thinner myelin sheaths compared with normally myelinated axons, an increased g-ratio indicates remyelination. PLP (proteolipid protein 1); MBP (myelin basic protein); MAG (myelin-associated glycoprotein); MOG (myelin oligodendrocyte glycoprotein); APC (adenomatous polyposis coli); GST (glutathionine S-transferase); LFB/PAS (luxol-fast-blue/periodic acid–Schiff reaction); APP (amyloid beta precursor protein). Scale bar 250 µm. Created with https://www.biorender.com at the 10 January 2024.

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
