# Peer review of "How to Use the Cuprizone Model to Study De- and Remyelination"

_ijms, 2024, doi:10.3390/ijms25031445_

Round 1

Reviewer 1 Report (New Reviewer)

Comments and Suggestions for Authors

This review is largely based on histological and anatomical parameters showing how the CPZ model can be used to further study demyelination and remyelination, and is a useful guide for those entering the field. However, one aspect that is lacking is comment on the behavioural/electrophysiological affects of CPZ treatment. Yes changes in myelination are observed but how does this relate to functional behaviour, an important consideration for translational studies to MS. 

The reference list needs attention for consistency and accuracy of citations.

Please see attachments for specific comments.

Comments on the Quality of English Language

Minor grammatical and linguistic changes needed.

Author Response

Reviewer 1:

We want to thank the reviewer for her/his efforts and time spent to review this manuscript.

Q1: Missing and incorrect references

A1: Thank you for this note, the references were updated (when possible) and new references were added.

Q2: Concentration of cuprizone.

A2: This information was added in the revised version of themanuscript.

Q3:  It is important to note that this demyelination is regional and not equal throughout the CNS. Most studies use the corpus callosum as the histological readout for demyelination but other CNS area show more/less demyelination. This needs to be recognised.

A3: Thank you for this note. I have added a statement accordingly. Beyond, it is not mentioned in the figure legend that the fornix is less vulnerable in this model

Q4: It should also be noted that just because there is evidence of demyelination this may have little or no effect on action potential propagation. Indeed there is very little behavioural evidence of symptoms that are commonly associated with human MS and electrophysiological studies in the CPZ are lacking.

A4: This is an excellent point, and I have addressed this in the conclusion section of this manuscript. In particular it is now stated:” In contrast to the EAE model, cuprizone-induced demyelination does not lead to severe motor deficits [116], limiting the evaluation of motor behavior as functional readout. While ultrastructural studies are the gold standard for evaluating myelin integrity and de- and remyelination, electrophysiological studies represent a powerful tool to assess the functional status of the axon-myelin unit [117], yet systematic studies addressing changes of electrophysiological properties following de- and remyelination in this model are largely missing. Crawfod and colleagues were able to demonstrate that cuprizone intoxication for three to six weeks induced significant demyelination in the corpus cal-losum, decreased axonal conduction, and caused deterioration in axonal structural in-tegrity, including at the nodes of Ranvier. Substituting the cuprizone diet with a regular diet resulted in myelin regeneration, however, it did not completely restore the deficits in conduction and structure [118]. A decrease in axonal conduction during cu-prizone-induced demyelination was as well found by our group [54]. Visual evoked po-tentials [119], a measurement frequently used during remyelination trials in humans, might be another promising additional readout not just in the EAE, but also in the cu-prizone model [105]. However, the extent of demyelination and axonal injury in the vis-ual pathway of cuprizone-intoxicated mice remains to be investigated. While there are proteomic alterations, clear demyelination was not seen in one study using a low dose (id est, 0.1%) cuprizone intoxication protocol [120].” The suggested reference was as well included.

Reviewer 2 Report (Previous Reviewer 2)

Comments and Suggestions for Authors

Authors have addressed all my comments. 

Author Response

Thank you for your time!

This manuscript is a resubmission of an earlier submission. The following is a list of the peer review reports and author responses from that submission.

Round 1

Reviewer 1 Report

Comments and Suggestions for Authors

The manuscript examines the cuprizone model, a crucial tool for investigating both demyelination and, notably, remyelination. It thoroughly explores experimental procedures and findings, emphasizing the model's significance, yet several key issues hinder its readiness for publication. Authored by Markus Kipp, who has previously collaborated with Sandra Amor's esteemed lab renowned for its contributions to demyelination research and animal models. Noteworthy, he has also recently contributed to the scientific area of the cuprizone model with an article titled "Astrocytes: Lessons Learned from the Cuprizone Model," published in November. The submitted manuscript though, requires significant revisions to enhance clarity, address methodological gaps, and improve language expression, rendering it currently unsuitable for publication.

Major comments:

Line 50: The statement "several different animal models exist..." should include the names of the other two prominent models, Experimental Autoimmune Encephalomyelitis (EAE) and Theiler's Murine Encephalomyelitis Virus (TMEV), for a comprehensive overview.

Line 56: Clarify the specific association between the pro-apoptotic transcription factor DNA damage-inducible transcript 3 (DDIT3) and endoplasmic reticulum stress responses.

Line 116: The abrupt transition at this point could benefit from either a separate section or an introduction, perhaps detailing the evaluation of oligodendrocyte loss.

Line 116: What compounds? What are these time points for? This is somehow confusing.

Figure 1. Some text is not discernible i.e the DAB immunohistochemistry. I also believe this protocol depiction is not necessary (plus the photos next to it are immunofluorescence-based). Additionally, the legend does not clearly describe all the panels in the figure. This is replicated in Figures 2 and 3, unfortunately. What I would suggest is to place an introductory schematic illustration demonstrating the cuprizone model instead. Not everyone is familiar with the first weeks of the animal feeding, nor you describe it anywhere in the manuscript. You cannot expect the reader to reach Line 284 to understand the model.

Line 130: anti-OLIG2 can also stain progenitor cells

Line 136: What about MOG or NOGO-A?

Line 180-186: This paragraph is very confusing and again here the figure does not help the reader to understand the message of the potential intervention.

Line 196: Which challenge? Sentence is not syntactically correct

Line 209: It’s not “studies” in plural; it is authors’ own sole “study” and it is also observational, not functional.

Line 231: Update the citation from J Histochem Cytochem 1990 to a more recent and relevant one related to Luxol Fast Blue.

Line 239: Repairing lost myelin sheaths IS NOT highly effective in MS. It can be in instances and subsets of MS patients, as Prof. Lassmann identified, however the process cannot be described as “effective”, rather present.

Line 253: What about the contribution of subgranular zone (SGZ), a narrow layer of cells located between the granule cell layer and hilus of the dentate gyrus? Are there any available data?

Line 261: Provide at least two credible sources, excluding the Xing study (ref 66), for OPCs distribution in normal white and grey matter of the CNS.

Line 296-301: This absolutely needs an illustrative figure. The time point described thereafter, should also be illustrated. This is probably the most interesting part of the manuscript and it is very poorly described.

Line 324: Please do the same as you did in Line 261 for clarity.

Line 333: Specify practical methods for manipulating the regions of interest. How can someone approach it experimentally?

Line 335: The conclusion is flat; adding nothing to the narrative of experimental manipulation. At least refer to scheduling, time points etc.

Comments on the Quality of English Language

The manuscript's English expression varies from mediocre to poor, occasionally resembling a suboptimal translation, as evidenced by phrases like "let the drug do its job" and "in our hands"

Author Response

Q1: Line 50: The statement "several different animal models exist..." should include the
names of the other two prominent models, Experimental Autoimmune Encephalomyelitis
(EAE) and Theiler's Murine Encephalomyelitis Virus (TMEV), for a comprehensive
overview.
A1: Thank you for this comment. We have added the information as suggested.
Q2: Line 56: Clarify the specific association between the pro-apoptotic transcription factor
DNA damage-inducible transcript 3 (DDIT3) and endoplasmic reticulum stress responses.
A2: Thank you for this comment. We have modified this part of the manuscript, explaining
more in detail the link between ER stress, unfolded protein response induction, and cellular
homeostasis. In particular, we now state: "From a functional point of view, myelinogenesis
and maintenance is a highly complex process that requires cell cycle exit and extensive
differentiation processes of oligodendrocyte progenitor cells. According to estimates from
morphometric analyses, the mean surface area of myelin per mature myelinating cell is
thousands of times greater than the surface area of a typical mammalian cell [47]. To form
and maintain the myelin sheath, oligodendrocytes synthesize large amounts of lipids and
proteins within the endoplasmic reticulum (ER), which increases their sensitivity to
perturbations of the secretory pathway. Due to the turnover of myelin lipids and proteins,
oligodendrocyte sensitivity to extrinsic and intrinsic disturbances persists till maturity.
Recent studies indicate that this extraordinary oligodendrocyte susceptibility might
contribute to the pathogenesis of a number of myelin disorders, including MS [47]. A
plethora of cell stress conditions can perturb the fine-tuned folding machinery within the
ER, a phenomenon which has been referred to as ER stress. As a consequence, cells
have evolved an adaptive coordinated response to limit accumulation of unfolded pro-teins
in the ER: the unfolded protein response (UPR). In higher eukaryotes, the UPR is activated
by three ER transmembrane sensors: Protein Kinase RNA-like Endoplasmic Reticulum
Kinase (PERK), activating transcription factor 6 (ATF6), and inosi-tol-requiring enzyme-1
(IRE1) [48]. Mechanistically, the UPR can relieve cell stress by (i) downregulating
translation and thus decreasing protein load within the ER-lumen, (ii) up-regulating genes
encoding ER chaperones and enzymes to increase the ER fold-ing capacity, and (iii)
inducing pathways to enhance degradation of misfolded proteins from the ER. However, if
cell stress persists and protective mechanisms fail, apoptosis is induced, among others, via
the transcription factor Chop (C/EBP homologous pro-tein) also known as DDIT3. Studies
using Ddit3-/- mice have established the role of DDIT3 during stress-induced apoptosis in a
number of disease models, including renal dysfunction [49], diabetes [50], ethanol-induced
hepatocyte injury [51], experimental colitis [52], advanced atherosclerosis [53] or cardiacpressure
overload [54]. Further-more, there is strong evidence that DDIT3 regulates cell
death in neurodegenerative and neuroinflammatory disorders, including Parkinson's
disease [55], subarachnoid hemorrhage [56], Alzheimer's disease [57], multiple system
Seite 3
atrophy [58], and spinal cord injury [59]. Notably, various ER stress markers are expressed
in inflammatory MS lesions [60, 61], as well as in experimental autoimmune
encephalomyelitis (EAE) [62, 63] and the cuprizone model [43]. Of note, oligodendrocyte
degeneration and demye-lination is ameliorated in Ddit3-/- compared to wildtype mice
[18]."
Q3: Line 116: The abrupt transition at this point could benefit from either a separate section
or an introduction, perhaps detailing the evaluation of oligodendrocyte loss.
A3: Thank you for this comment; we have tried to smoothen the transition by addressing
the relevance and strategies to evaluate oligodendrocyte loss. In particular, we now state,
"One promising strategy to ameliorate MS disease progression is the protection of
oligodendrocytes. Due to the selective degeneration of these cells during early cu-prizone
intoxication, this model is particularly suitable to study pathways and drugs able to
ameliorate oligodendrocyte death. In this section points are discussed which might be
considered during the experimental design of a study addressing this point."
Q4. Line 116: What compounds? What are these time points for? This is somehow
confusing.
A4: The term "compounds" was replaced with "drugs".To study oligodendrocyte loss in the
cuprizone model, early time points should be investigated, and we suggest analyzing
oligodendrocyte loss after 3 and 7 days post initiation of the cuprizone intoxication. We
have modified these sections with the aim of increasing the readability and clarity.
Q5: Figure 1. Some text is not discernible i.e the DAB immunohistochemistry. I also believe
this protocol depiction is not necessary (plus the photos next to it are immunofluorescencebased).
Additionally, the legend does not clearly describe all the panels in the figure. This
is replicated in Figures 2 and 3, unfortunately. What I would suggest is to place an
introductory schematic illustration demonstrating the cuprizone model instead. Not
everyone is familiar with the first weeks of the animal feeding, nor you describe it anywhere
in the manuscript. You cannot expect the reader to reach Line 284 to understand the
model.
A5: Following the suggestion of the kind reviewer, we now place an introductory schematic
illustration demonstrating the cuprizone model and address this part early in the
manuscript. The content of the original figures was significantly reduced, and the figure
legends were adopted accordingly.
Q6: Line 130: anti-OLIG2 can also stain progenitor cells
A6: Thank you for this comment. We now state the following: “One should note that the
transcription factor OLIG2 is expressed by both, oligoden-drocyte progenitor cells and
mature oligodendrocytes. Single labelling with an-ti-OLIG2 antibodies, thus, provides
limited information about mature oligodendrocyte densities.”
Q7: Line 136: What about MOG or NOGO-A?
A7: While NOGO-A would also be an appropriate marker to label oligodendrocyte cell
bodies, oligodendrocyte cell bodies are hard to separate in anti-MOG stained sections.
Thus, we do not recommend this stain to evaluate the extent of oligodendrocyte
degeneration in the cuprizone model.
Q8: Line 180-186: This paragraph is very confusing and again here the figure does not
help the reader to understand the message of the potential intervention.
A8: We have modified both the paragraph and illustration. In particular, we now state,
"Some important aspects should be considered when using the cuprizone model to
Seite 4
investigate the potentially protective properties of substances on demyelination. First,….
And later on.. Second, potential protective drug effects are best studied using an
experimental setting where demyelination is semimaximal and not complete."
Q9: Line 196: Which challenge? Sentence is not syntactically correct
A9: Thank you for this comment. We have adopted the section. In particular, we now state,
"In our opinion, at least three different strategies might be applied to optimize the
experimental conditions."
Q10: Line 209: It’s not “studies” in plural; it is authors’ own sole “study” and it is also
observational, not functional.
A10: We adopted the section of the manuscript accordingly.
Q11: Line 231: Update the citation from J Histochem Cytochem 1990 to a more recent and
relevant one related to Luxol Fast Blue.
A11: Thank you for this comment. We have added more citations, including recent ones.
Q12: Line 239: Repairing lost myelin sheaths IS NOT highly effective in MS. It can be in
instances and subsets of MS patients, as Prof. Lassmann identified, however the process
cannot be described as “effective”, rather present.
A12: Thank you for this comment, highly appreciated. We clarified this aspect. In particular,
we now state, "While the regeneration of destroyed neurons and their axons in the adult
CNS is highly limited, repairing lost myelin sheaths is principally possible. About 20-30% of
examined postmortem MS tissues show signs of remyelination, which can happen at both
early and late stages of the disease [75, 76]".
Q13: Line 253: What about the contribution of subgranular zone (SGZ), a narrow layer of
cells located between the granule cell layer and hilus of the dentate gyrus? Are there any
available data?
A13: Thank you for this comment. In contrast to SVZ-derived stem cells, SGZ-derived stem
cells mainly differentiate into neurons unless they are genetically reprogrammed and/or
trophically manipulated to produce oligodendrocytes (Cells. 2019 Aug; 8(8): 825). For the
interested reader, we now cite this recent review article. Beyond, we further list the median
eminence as a potential source of stem cells with the potential to differentiate into
oligodendrocytes (https://pubmed.ncbi.nlm.nih.gov/32413277/)
Q14: Line 261: Provide at least two credible sources, excluding the Xing study (ref 66), for
OPCs distribution in normal white and grey matter of the CNS.
A 14: Thank you for this comment. As suggested, we provide the following citations for the
statement that OPCs are widely distributed in the normal white and grey matter of the
CNS. https://pubmed.ncbi.nlm.nih.gov/14572468/
https://pubmed.ncbi.nlm.nih.gov/12617946/
https://pubmed.ncbi.nlm.nih.gov/24744380/
https://pubmed.ncbi.nlm.nih.gov/30503207/
Q15: Line 296-301: This absolutely needs an illustrative figure. The time point described
thereafter should also be illustrated. This is probably the most interesting part of the
manuscript, and it is very poorly described.
A15: As suggested by the reviewer, we discuss this aspect of the cuprizone model more
extensively in the revised version of the manuscript. Beyond this, figure 1 illustrates the
limited remyelination potential after chronic cuprizone-induced demyelination. In particular,
Seite 5
we now state the following: "During a 12 weeks continuous cuprizone intoxicaton protocol it
has been shown that GSTπ+ mature oligodendrocyte densities are significantly reduced till
week 5 but then recover around week 6. Between week 6 and 12, these maturating
oligodendrocytes become again vulnerable against the cuprizone intoxication resulting in a
second wave of oligodendrocyte apoptosis, detected via the TUNEL assay [27]. This
second wave of mature oligodendrocyte apoptosis was paralleled by a progressive loss of
NG2+ pro-genitors within the chronically demyelinated corpus callosum which might
explain the impaired endogenous remyelination capacity after chronic cuprizone-induced
demye-lination. In fact, long-term feeding of cuprizone induces an episodic sequence of
de-myelinating and remyelinating events which becomes ineffective over time, progressing
to a chronic state of demyelination [16]. The relevance of axonal injury for remyelination
failure is controversially discussed. Mason and colleagues rarely observed ne-crotic axons
but showed a mean decrease in axonal caliber indicating some kind of ax-onal injury or
dysfunction which might impair remyelination [16]. In line with this observation, we and
others showed a disturbed axonal transport machinery in cu-prizone intoxicated mice,
evidenced by the accumulation of pre-synaptic proteins and/or mitochondria [28, 98-101].
Alternatively, alterations in the growth factor environment, which is not permissive for
oligodendrocyte progenitor proliferation and differentiation after chronic cuprizone-induced
demyelination, might as well play a role".
Q16: Line 324: Please do the same as you did in Line 261 for clarity.
A16: Again, we added the respective citations at this point.
Q17: Line 333: Specify practical methods for manipulating the regions of interest. How can
someone approach it experimentally?
A17: Thank you for this comment. We have modified these sections accordingly. In
particular, we now state the following: "Keeping this in mind, if substances are used that
are expected to modulate NPC-derived oligodendrocytes rather than pOPC, such as Gli1
inhibitors [106], the fo-cus of the region of interest should be at rostral rather than caudal
levels of the corpus callosum to be able to measure the potential pro-myelinating effects of
NPC-derived oligodendrocytes.".
Q18: Line 335: The conclusion is flat, adding nothing to the narrative of experimental
manipulation. At least refer to scheduling, time points etc.
A18: Following this final suggestion, we have substantially modified the final conclusion of
the manuscript.

Reviewer 2 Report

Comments and Suggestions for Authors

Review

Kipp et al. reviewed the current understanding of cuprizone model and its applications in preventing oligodendrocyte degeneration and/or enhancing remyelination.. This paper addressed a significant clinical problem, and it is well-written. I enjoy reading this manuscript.

Following are my comment points.

Major points:

1.     For section 2, the idea of cuproptosis should be incorporated, given that cuprizone-induced copper dysregulation contributes to the oligodendrocytes degeneration.

2.     For functional studies about oligodendrocyte and demyelination, it would be helpful to comment on electrophysiological parameters and cellular activity assays.

Author Response

Q1: For section 2, the idea of cuproptosis should be incorporated, given that cuprizoneinduced
copper dysregulation contributes to the oligodendrocytes degeneration.
A2: Thank you for reviewing this manuscript and your valuable feedback. We have
incorporated both points in the final conclusion sections of the manuscript.
Q2: For functional studies about oligodendrocyte and demyelination, it would be helpful to
comment on electrophysiological parameters and cellular activity assays.
A2: We have incorporated both points in the final conclusion sections of the manuscript.

Round 2

Reviewer 1 Report

Comments and Suggestions for Authors

The primary directive was to restructure the paper into a consistent, reader-friendly format suitable for those unfamiliar with the cuprizone model—a review guide, per se, given the title: “How to use the cuprizone model to study...”. Regrettably, the current state of the manuscript still falls short of the necessary standards.

1. Structural Inconsistencies: The implemented structural changes, intended to enhance consistency and readability, are deemed suboptimal. The figures, specifically Figure 2 indicating durations of 3 days and 7 days for apoptotic cells, alongside new Figure 1 illustrating apoptosis/ferroptosis after week 2, lack coherence. Similarly, Figure 3 referencing five and three weeks of intoxication, followed by Figure 4 depicting five weeks only, and subsequent 1-2 weeks of normal chow, lack a unified timeline. This inconsistency confounds rather than elucidates crucial timepoints. This was highlighted in the initial assessment.

2. Schematic Illustration Deficiencies: Despite the addition of a schematic, crucial details are omitted. There is a pronounced absence of indications regarding cuprizone or standard chow exchange, information about de- or remyelination, and insights into potential drug interventions. The myelination level alone lacks context. Additionally, the font size compromises legibility, further diminishing the efficacy of the illustration. The concern escalates given the blatant reuse of this schematic in a previous review published in the same journal in December 2022 (Remyelination in Multiple Sclerosis: Findings in the Cuprizone Model). This raises questions about the originality and significance of the present work, especially considering a similar review from 2020 in Cells (The Cuprizone Model: Dos and Do Nots).

3. Omission of Key Term: Notably, the term "g-ratio," pivotal in discussions of one of the primary animal models of remyelination, remains conspicuously absent. Its exclusion diminishes the article's qualification for publication, on top of what already has been mentioned.

Comments on the Quality of English Language

English language still indicate room for improvement.

Author Response

Reviewer 1:

We want to thank the reviewer for her/his efforts and time spent to review this manuscript. We realize the kind reviewer is not convinced about this little work, which we regret.

Q1: Structural Inconsistencies: The implemented structural changes, intended to enhance consistency and readability, are deemed suboptimal. The figures, specifically Figure 2 indicating durations of 3 and 7 days for apoptotic cells, alongside new Figure 1 illustrating apoptosis/ferroptosis after week 2, lack coherence. Similarly, Figure 3 referencing five and three weeks of intoxication, followed by Figure 4 depicting five weeks only, and subsequent 1-2 weeks of normal chow, lack a unified timeline. This inconsistency confounds rather than elucidates crucial timepoints. This was highlighted in the initial assessment.

A1: Thank you very much for your comments. Figure 1 illustrates cellular key events during cuprizone-induced demyelination and subsequent remyelination, as stated in the figure legend. These include the degeneration of oligodendrocytes (here, apoptosis and ferroptosis), microglia activation, myelin disintegration, and the activation and differentiation of OPCs. All subsequent figures aim to illustrate a typical experimental setup to study the potential effectiveness of drugs to ameliorate oligodendrocyte degeneration, to ameliorate cuprizone-induced demyelination, or to study the potential effectiveness of drugs to support remyelination. Again, this is clearly stated in the respective figure legends.

The different timings of the analyses result from the different research questions. Oligodendrocyte degeneration is an early event during cuprizone intoxication. Therefore, we recommend analyzing two early time points (i.e., 3 and 7 days, see Figure 2). On the other hand, if the pharmacological influence of a substance on demyelination is to be examined, we suggest a cuprizone intoxication duration of either 3 or 5 weeks (see Figure 3). Finally, Figure 4 illustrates an often used experimental design in which a 5-week cuprizone intoxication induces acute demyelination, and then after 7 and 14 days the myelin status is determined during endogenous remyelination. Unfortunately, we have not succeeded in clearly explaining this critical aspect. Therefore, we have decided to adjust the figures accordingly and address this critical point in the text. In particular, we now state: “At least three different strategies might be applied to optimize the experimental conditions. First, one might focus on those regions where demyelination is incomplete, such as the midline of the corpus callosum genu. This is illustrated in Figure 3A. While after five weeks of cuprizone intoxication, the midline of the corpus callosum trunk is almost completely demyelinated (right anti-PLP stained images), demyelination is incomplete in the midline of the corpus callosum genu (left anti-PLP stained images). Second, one might terminate the experiment after three weeks and analyze the tissue while myelin damage and removal are ongoing and not at their peak. This is illustrated in Figure 3B. While myelin disintegration is visible in LFB/PAS stained sections, anti-PLP staining intensity loss is less obvious. Third, one might rely on the observation that three weeks of cuprizone intoxication is enough to allow autonomous lesion progression till week five (not illustrated in figure 3)”.

Q2: Schematic Illustration Deficiencies: Despite the addition of a schematic, crucial details are omitted. There is a pronounced absence of indications regarding cuprizone or standard chow exchange, information about de- or remyelination, and insights into potential drug interventions. The myelination level alone lacks context. Additionally, the font size compromises legibility, further diminishing the efficacy of the illustration. The concern escalates given the blatant reuse of this schematic in a previous review published in the same journal in December 2022 (Remyelination in Multiple Sclerosis: Findings in the Cuprizone Model). This raises questions about the originality and significance of the present work, especially considering a similar review from 2020 in Cells (The Cuprizone Model: Dos and Do Nots).

A2: The font size was adopted, as suggested. Indeed, our group has published several articles using the cuprizone model, and we have contributed to the scientific field with various review articles, starting with a Review article in 2009, published in Acta Neuropathologica (The cuprizone animal model: new insights into an old story). While the two review articles address the same topic, they focus on different aspects of the model. In the  December 2022 article, we entirely focused on recent findings using the cuprizone model in the context of remyelination and discussed there the potential of some of the identified compounds to promote remyelination in multiple sclerosis patients. Indeed, the current illustration two was adopted from that one. To clarify this point, we clearly state this in the revised version of the manuscript.

Regarding the second article mentioned by the reviewer (The Cuprizone Model: Dos and Do Nots), we would like to point out that several new aspects are discussed in the current work, including the process of ferroptosis, the source of remyelinating oligodendrocytes, recent findings using S1PR modulation, etc. In our opinion, the current article represents a significant contribution to the field. However, it is very fine for us to agree that we disagree.

Q3: Omission of Key Term: Notably, the term "g-ratio," pivotal in discussions of one of the primary animal models of remyelination, remains conspicuously absent. Its exclusion diminishes the article's qualification for publication, on top of what already has been mentioned.

A3: Thank you for this comment; we mention the definition and relevance of the g-ratio in the revised version of the manuscript. In particular, we now state: Further note that by using electron microscopy, a so-called ‘g-ratio’ can be calculated by dividing the diameter of the axon by the diameter of the axon+myelin sheath. Since remyelinated fibers are characterized by thinner myelin sheaths compared to normally myelinated axons, an increased g-ratio indicates remyelination.

Round 3

Reviewer 1 Report

Comments and Suggestions for Authors

Thank you for the revised submission. Unfortunately, the figures remain unclear and do not enhance understanding. Adding a sentence or two falls short of securing a comprehensive review on remyelination. The current additions do not ensure a comprehensive review of how to use the cuprizone model to study de- and remyelination. A single, clear, and illustrative figure, which was requested from the first revision, is crucial for readability.

Comments on the Quality of English Language

English language still indicate room for improvement.